# Assessing Smooth Pursuit Eye Movements Using Eye-Tracking Technology in Patients with Schizophrenia Under Treatment: A Pilot Study

**DOI:** 10.3390/s25165212

**Published:** 2025-08-21

**Authors:** Luis Benigno Contreras-Chávez, Valdemar Emigdio Arce-Guevara, Luis Fernando Guerrero, Alfonso Alba, Miguel G. Ramírez-Elías, Edgar Roman Arce-Santana, Victor Hugo Mendez-Garcia, Jorge Jimenez-Cruz, Anna Maria Maddalena Bianchi, Martin O. Mendez

**Affiliations:** 1CI3M Lab, Facultad de Ciencias, Universidad Autónoma de San Luis Potosí, San Luis Potosí 78295, Mexico; l.b.c.c@outlook.com (L.B.C.-C.); groluis@hotmail.com (L.F.G.); fac@fc.uaslp.mx (A.A.); miguel.ghebre@uaslp.mx (M.G.R.-E.); arce@fciencias.uaslp.mx (E.R.A.-S.); victor.mendez@uaslp.mx (V.H.M.-G.); 2Department of Obstetrics and Prenatal Medicine, Hospital University of Bonn, 53127 Bonn, Germany; jorge.jimenez_cruz@ukbonn.de; 3Department of Electronics, Information and Bioengineering, Politecnico di Milano, 20133 Milan, Italy; annamaria.bianchi@polimi.it; 4Fondazione IRCCS Ca’ Granda Ospedale Maggiore Policlinico, 20122 Milan, Italy

**Keywords:** SVM, trajectory analysis, eye movement, classification, psychiatric disorder

## Abstract

Schizophrenia is a complex disorder that affects mental organization and cognitive functions, including concentration and memory. One notable manifestation of cognitive changes in schizophrenia is a diminished ability to scan and perform tasks related to visual inspection. From the three evaluable aspects of the ocular movements (saccadic, smooth pursuit, and fixation) in particular, smooth pursuit eye movement (SPEM) involves the tracking of slow moving objects and is closely related to attention, visual memory, and processing speed. However, evaluating smooth pursuit in clinical settings is challenging due to the technical complexities of detecting these movements, resulting in limited research and clinical application. This pilot study investigates whether the quantitative metrics derived from eye-tracking data can distinguish between patients with schizophrenia under treatment and healthy controls. The study included nine healthy participants and nine individuals receiving treatment for schizophrenia. Gaze trajectories were recorded using an eye tracker during a controlled visual tracking task performed during a clinical visit. Spatiotemporal analysis of gaze trajectories was performed by evaluating three different features: polygonal area, colocalities, and direction difference. Subsequently, a support vector machine (SVM) was used to assess the separability between healthy individuals and those with schizophrenia based on the identified gaze trajectory features. The results show statistically significant differences between the control and subjects with schizophrenia for all the computed indexes (*p* < 0.05) and a high separability achieving around 90% of accuracy, sensitivity, and specificity. The results suggest the potential development of a valuable clinical tool for the evaluation of SPEM, offering utility in clinics to assess the efficacy of therapeutic interventions in individuals with schizophrenia.

## 1. Introduction

Schizophrenia (SZ) is a complex psychiatric disorder marked by cognitive dysfunctions including impairments in attention, memory, and processing speed [1] affecting cognition, behavior, affectivity, and sensory-perception, which can lead to an emotional and social imbalance of the individual [2]. According to the World Health Organization, SZ represents a serious health problem, affecting approximately 0.3% to 0.7% of the global population [3].

Eye movements have different behaviors depending on the scene or stimuli received. These are fixation, saccadic movement, and smooth pursuit. Fixation movement occurs when the gaze remains static; this is characterized by having a speed < 30 deg/s, with respect to the center of rotation of the eye, and a duration of at least 200 ms. It is common to define areas of interest in order to measure a fixation occurrence when static stimuli are present. Smooth pursuit is the eye movement when observing a moving target. Its function is to stabilize the image so that the eye is able to focus on the object despite its movement; its speed and duration depend on the objective. Saccade movements are those movements made by the eye to quickly focus on an objective and occur between fixations or in the middle of smooth pursuit; their duration is between 30 and 80 ms and speed is between 30 and 500 deg/s, and they occur when a new element emerges or to explore the scene [4,5,6,7].

The diagnosis of SZ is based on the detection of a minimum number of key alterations in mental function [8] divided into cognitive domains: processing speed, attention/vigilance, working memory, social cognition, reasoning and problem solving, learning, and visual memory. In severe forms, the evident alterations allow for a faster diagnosis. However, detecting SZ in its early stages is challenging as symptoms may not be clear and an objective, quantitative evaluation strategy is lacking. Additionally regular assessments after diagnosis are crucial to evaluate the patient’s evolution under medication. This evaluation demands expert personnel employing quantitative measures to determine the therapy outcome. Therefore, finding innovative strategies to aid physicians in diagnosing, following-up, and detecting early stages of SZ is imperative.

Previously published works have shown that individuals with SZ exhibit significant alterations in various domains of visual processing, particularly in perceptual organization and motion detection, which are consistently impaired in this population [4,5,9]. These visual processing deficits are not isolated phenomena but are closely intertwined with broader cognitive dysfunctions. For instance, impairments in perceptual organization have been linked to deficits in attention and working memory [10,11]. At the same time, abnormalities in eye movements, particularly in smooth pursuit, reflect underlying disruptions in processing speed and visual memory [7,12]. Since gaze control is a cognitive-motor function, disruptions in visual tracking tasks can serve as indirect markers of attentional lapses and slowed cognitive processing [2,6]. This underscores the importance of studying smooth pursuit eye movements to assess the cognitive impairment characteristics of SZ, specifically those related to attention, processing speed, and visual memory. In addition, patients with SZ consistently show increased saccade frequency, prolonged fixation durations, and reduced reading fluency, reflecting both oculomotor and higher-level semantic processing deficits. Furthermore, individuals with SZ show impaired top-down attentional control, leading to increased fixations and revisits to salient but irrelevant stimuli and difficulty suppressing distractors [13,14]. In particular, abnormalities in smooth pursuit have been shown to serve as indicators of psychosis in disorders, such as schizophrenia, schizoaffective disorder, and psychotic bipolar disorder [15,16,17,18]. While previous studies have examined eye movement abnormalities in schizophrenia, such research has largely been restricted to controlled experimental settings due to the challenge of obtaining reliable gaze data. Smooth pursuit eye movements, in particular, are recognized as potential biomarkers of schizophrenia, yet their clinical use has been limited by the technical difficulties in accurately recording and analyzing these motions.

Recent studies have increasingly employed eye-tracking technology to investigate smooth pursuit eye movement (SPEM) abnormalities in schizophrenia, highlighting its potential as a biomarker for the disorder. For instance, Ales et al. demonstrated that individuals instructed to feign schizophrenia could not replicate the SPEM deficits observed in actual patients, underscoring the specificity of these oculomotor anomalies to the disorder [19]. Similarly, Komogortsev and Karpov developed a ternary classification system to distinguish between fixations, saccades, and smooth pursuits, demonstrating the feasibility of automated SPEM analysis in clinical populations [20]. These findings align with our study’s approach, which leverages quantitative gaze trajectory features to distinguish between medicated patients with schizophrenia and healthy controls. Our methodology builds upon this foundation by integrating machine learning classification, offering a novel contribution to the growing body of literature on SPEM-based diagnostics. Furthermore, recently published reviews have highlighted the diagnostic potential of smooth pursuit eye movements (SPEMs). Lima and Ventura reviewed psychophysical eye-tracking designs and emphasized the utility of smooth pursuit metrics in assessing perceptual and cognitive deficits [21]. Startsev et al. reviewed eye movement detection research, suggesting that SPEM abnormalities may serve as reliable indicators of cognitive dysfunction and disease progression [22].

The goal of our study is to bridge the gap of eye movement tracking in clinical settings by conducting a comparative analysis of smooth pursuit eye movements in medicated individuals with schizophrenia versus healthy controls, using a visual tracking task. We aim to demonstrate that quantitative metrics (specifically trajectory-based measures such as colocalities, directionality, and polygonal area) can be used effectively with modern eye-tracking technology. Furthermore, by developing a support vector machine (SVM) classifier, we seek to establish a framework for an objective, clinically valuable tool to assess gaze behavior and aid in the diagnosis and monitoring of schizophrenia.

## 2. Material and Methods

### 2.1. Participants

Eighteen participants ranging in age from 19 to 54 years were enrolled at the psychiatric clinic of Dr. Everardo Neumann in San Luis Potosí, Mexico. The first group included nine individuals diagnosed with paranoid schizophrenia (SZ group, age 24.7 ± 7.5 years), undergoing treatment with atypical antipsychotics (risperidone 2 mg), olanzapine (10–15 mg), and quetiapine (350–650 mg) for at least 2 weeks; in the second group, there were nine controls (CNTs, age 22 ± 3.2 years), previously classified as healthy by a psychiatrist. The study protocol received approval from the local ethical committee of the psychiatric clinic and adhered to the principles outlined in the Declaration of Helsinki (1964). All the participants provided written informed consent. The participants with schizophrenia were clinically stable and receiving constant psychopharmacological treatment.

### 2.2. Experimental Setup

The eye-tracking data were collected using the EyeTribe Tracker (Developer Kit, Model 2016), manufactured by The Eye Tribe (Copenhagen, Denmark). The device is a low-cost, remote eye tracker operating at a sampling frequency of 30 to 60 Hz, with a spatial accuracy of approximately 0.5deg of visual angle. However, the acquired data showed an effective sampling frequency of approximately 20 Hz. This approximation accounts for the implicit latency introduced by both the device and the software during data storage. The resultant data was not equally spaced temporally; hence, it was resampled to 60 Hz by linear interpolation. The development environment was Processing version 3 [23] using the “Eye Tribe for Processing” library, and the post-processing data analysis was performed with MATLAB 2019 commercial software.

The stimulus was displayed on a 14” screen (1366 × 768 pixel resolution, Lenovo ideapad s400u©, made in Tierra del Fuego, Argentina), and the participants were seated 50 cm away using a chin rest (see Figure 1). Data recording, processing, and analysis were conducted on a computer with an I5–3337u processor and 4GB RAM. All the experiments were overseen by a psychiatrist at the hospital, and a custom-designed software using Processing 3 facilitated the test.

### 2.3. Calibration

The calibration quality was assessed for each participant based on the ability to detect stable and accurate gaze fixation on each of the seven calibration points. The calibration sequence included three points at the top, three at the bottom, and one in the center of the screen. The participants were instructed to fixate on each point until it disappeared. We verified that the gaze data aligned with the intended fixation targets before proceeding. If the calibration was unsuccessful—indicated by inconsistent or inaccurate fixation data—the process was repeated until a satisfactory level of tracking stability and spatial accuracy was achieved. Some participants, not included in the final group of nine analyzed in this study, were excluded for failing to meet the calibration criteria for stable and accurate gaze fixation after repeated attempts or for being unable to complete the tracking task.

### 2.4. Task

The participants tracked a white ball (target) with rectilinear motion at a speed of 250 pixels per second, moving against a black background screen for a duration of 14 s. The trajectory of the target, as illustrated in Figure 2, was designed with five stages according to the following steps:The target appears in the scene and remains static for one second.A random direction is selected, and the target moves in a straight line for five seconds, bouncing off the edges if encountered.The target comes to a stop for two seconds.Subsequently, the target resumes movement in the same direction for an additional five seconds, bouncing off the edges if encountered.Finally, it comes to a stop for one second before disappearing.

The moving ball test serves to evoke cognitive functions, including those associated with attention such as centering (maintaining and sustaining), directing (following), and attending (prioritizing) [24]. While it is possible that individuals with schizophrenia can perform attending functions, they may encounter challenges with centering functions [24] and directing attention [25]. The dynamic nature of the target in motion, including stops, facilitates the evaluation of visual memory and perseverance—the ability to sustain a specific behavior in response to stimuli [25]. To assess the cognitive status, spatiotemporal indices were calculated to capture the temporal dynamics of the trajectories and evaluate the deviation of the gaze trajectory from the target trajectory. In this work, the gaze behavior of the control subjects and those with SZ was studied during the SPEM trajectory. Therefore, ball and gaze trajectories were compared using colocalities, directions, and polygon area indices.

### 2.5. Trajectory Analysis

A trajectory is the path through which an object travels in space. Mathematically, it can be represented as a vector function $r\left(t\right)=\left(x_{t},y_{t}\right)$ which assigns a point in space to each time value *t*. In this work, the space consists of the two-dimensional rectangular lattice of all the pixel coordinates in the computer screen (i.e., $Z\times Z=Z^{2}$), and the time variable *t* is also discrete. Also, all the trajectories have been resampled to a sampling frequency of 60 Hz and have a fixed length *N* = 840, corresponding to a 14-second task, as described in the previous section. Therefore, a trajectory will be represented as follows:(1)$$r\left(t\right)=\left(x_{t},y_{t}\right),$$
where the ordered pair $\left(x_{t},y_{t}\right)$ represents the position of the object at time *t*, with t∈1,…,N and $\left(x_{t},y_{t}\right)\in Z^{2}$. Based on this definition, in the following subsections we describe the measures to quantify the relation between the target trajectory $r_{o}\left(t\right)$ and the gaze trajectory $r_{g}\left(t\right)$.

#### 2.5.1. Spatiotemporal Colocalities

Colocalities allow for the measurement of similarity between target and gaze trajectories by assessing the overlap of geometric figures. In this study, we utilized balls of different sizes to measure the overlap between the target and gaze balls at each time. Spatiotemporal colocalities represent a measure of similarity between the target $r_{o}\left(t\right)$ and gaze $r_{g}\left(t\right)$ trajectories. Given a metric *d* (e.g., the Euclidean distance), let us define a closed ball $B_{r}\left(x,y\right)$ as the set of all the lattices whose distance to (x,y) is at most *r* (here, *r* is the radius of the ball) [26]. This is expressed as(2)$$B_{r}\left(x,y\right)=\left\{\left(x^{\prime },y^{\prime }\right)\in Z^{2}|d\left(\left(x^{\prime },y^{\prime }\right),\left(x,y\right)\right)\leq r\right\}.$$

The spatiotemporal colocalities between the target and gaze trajectories can then be computed using the following formula:(3)$$Co=\frac{\sum_{t=1}^{t=N}\left|B_{r1}\left(r_{o}\left(t\right)\right)\cap B_{r2}\left(r_{g}\left(t\right)\right)\right|}{\sum_{t=1}^{t=N}\left|B_{r2}\left(r_{g}\left(t\right)\right)\right|},$$
where |.| is the cardinality of a set, *N* is the number of points in the trajectory, and r1 and r2 refer to radii. The colocality measure Co approaches one when the subject’s gaze closely follows the object’s trajectory at a distance between centers of at most r1+r2. However, even for healthy subjects, systematic deviations in $r_{g}\left(t\right)$ relative to $r_{o}\left(t\right)$ may arise due to system acquisition errors or gaze-tracking inaccuracies. To address this issue, different radii for $B_{r2}\left(r_{g}\left(t\right)\right)$ and $B_{r1}\left(r_{o}\left(t\right)\right)$ were tested. The radii were selected from the set of possible values: r1∈ {5 mm, 10 mm, and 15 mm} and r2∈ {2 mm, 4 mm, and 8 mm}.

The choice of these specific radii is based on visual inspection to ensure that the chosen values provide meaningful spatial information, since for very small radii (r1 and r2) the balls are too small and do not touch, which does not capture the interaction between them. For very large radii, the balls always overlap significantly, making it difficult to distinguish individual elements. So, these radii were chosen to balance the visual clarity between not touching and excessive ball overlap. Finally, it is important to note that failure to follow the moving target accurately could suggest difficulties in directing and maintaining attention.

#### 2.5.2. Direction

As the object moves in a straight line, it is expected that the gaze will follow a similar path, that is, with constant direction. Since the direction of the subject’s gaze trajectory $r_{g}\left(t\right)$ is expected to be similar to the trajectory of the moving object $r_{o}\left(t\right)$, we evaluate how closely the direction of $r_{g}\left(t\right)$ matches that of $r_{o}\left(t\right)$ at each time point *t*. To achieve this, we compute for each trajectory the angle θ(t) between the line segment connecting two consecutive points and the positive *x*-axis with the four-quadrant arctan function. That is, given two consecutive points ($x_{t}$,$y_{t}$) and ($x_{t+1},y_{t+1}$) of a trajectory, the angle θ(t) is $\theta \left(t\right)=arctan2(y_{t+1}-y_{t},x_{t+1}-x_{t}$) for *t* = 1, …, *N* - 1.

To quantify the similarity between the gaze and object trajectories, we computed the circular difference between the angles of the two trajectories [27] as(4)$$\Delta \theta_{sample}=1-\left|\frac{1}{N}\sum_{t=1}^{N}\exp^{i(\theta_{o}\left(t\right)-\theta_{g}\left(t\right))}\right|,$$
where *N* is the number of samples, $\theta_{o}$(*t*) is the angle for the object trajectory, and $\theta_{g}$(*t*) is the angle for the gaze trajectory at each time. Note that (a) if the phases $\theta_{o}$(*t*) and $\theta_{g}$(*t*) are perfectly aligned over time, the exponential terms will consistently point in the same direction on the unit circle, and the magnitude will be close to one, and (b) if the phases are randomly distributed, the mean will tend to cancel and the magnitude will approach zero. So, a Δθ value close to zero indicates high phase coherence, while a value close to one indicates phase variability.

The direction index was computed in two ways: (a) using consecutive samples to evaluate the direction at each sample ($\Delta \theta_{sample}$) and (b) identifying the bouncing points of the objective ball, computing the regression line of the points closed by consecutive bouncing points, and obtaining the angle of the regression line ($\Delta \theta_{regr}$). This means, for straight segments of the trajectories, we performed a linear regression on the points corresponding to the gaze trajectory. The angle of the regression line from the angle of the target trajectory was subtracted and averaged these angle differences across all straight segments for each recording (here called $\Delta \theta_{regr}$).

### 2.6. Area of Polygons

The trajectory of the object consists of straight-line segments. For each such segment, if a polygon is formed by *n* successive points and its area *A* is computed, the expected result is a value close to zero. The same is true for the gaze trajectory, assuming that the gaze follows a nearly straight path. However, if the person deviates from the task, the trajectory may no longer be a straight line, resulting in a polygon with a non-zero area. With this motivation, the average area of the polygons formed by the trajectories was calculated in non-overlapping segments, with *n* samples ranging from 15 to 90.

The Shoelace method was used to calculate the area of these polygons [28]. This method is particularly efficient at calculating the area of a polygon using its vertex coordinates. It accurately handles both simple and complex polygons (those with self-intersections) by summing the contributions of each vertex pair. The Shoelace method works by calculating the cross products of successive pairs of vertex. For a polygon with vertices ($x_{1},y_{1}$), ($x_{2},y_{2}$), …, ($x_{n},y_{n}$), the area *A* is given by(5)$$A=\frac{1}{2}\left|\sum_{i=1}^{n}\left(x_{i}y_{i+1}-y_{i}x_{i+1}\right)\right|,$$
where ($x_{n+1},y_{n+1}$) = ($x_{1},y_{1}$). In the case of complex polygons, the Shoelace method inherently accounts for these intersections by summing the signed areas, taking into account the orientation (clockwise or counterclockwise) of the vertices. This ensures an accurate calculation of the total area, even if the polygon is not simple.

## 3. Statistics

The statistical differences between the trajectory measures (colocalities, direction, and area of polygons) of the control and SZ samples were computed. The first step was to verify the non-normality of the samples; thus, the non-parametric Kolmogorov–Smirnov test (KS) was applied to the trajectory measures for each group. The KS test evaluates the differences in two different cumulative distributions on the basis of the maximum distance between them. The normality test was rejected. Since the normality test was rejected, the Wilcoxon–Mann–Whitney test was applied to evaluate the hypothesis that the two samples originate from the same population.

To quantify the magnitude of group differences, we computed Hedges’ g, a standardized effect size measure that adjusts for small-sample bias. Hedges’ g expresses the mean difference between two groups in terms of pooled standard deviation units, offering a scale-invariant index of effect magnitude. Values around 0.2 are interpreted as small, around 0.5 as medium, and 0.8 or higher as large effects. In our analysis, Hedges’ g was used alongside *p*-values to provide a more comprehensive understanding of the practical significance of the observed differences.

## 4. Results

The gaze trajectories were compared using colocalities, directions, and polygon area indices. Table 1 shows the mean values of the colocalities, direction, and area of polygon indices for each participant. An asterisk (*) indicates a statistically significant difference between the groups for a given parameter.

### 4.1. Colocalities

When evaluating colocalities, the controls showed significantly higher overlap with the target paths (mean = 0.98) than the patients with SZ (mean = 0.87); *p*< 0.01. Figure 3 displays the trajectories of the balls, where the target ball (filled gray) has a radius of r1 = 11mm and the gazing ball (empty black) has a diameter of r2 = 5.5 mm. Top panel displays the ball trajectories for the subjects with SZ, while bottom panel displays the ball trajectories for the control subjects. As shown in Figure 3, the gaze trajectory ball of the control subjects tends to overlap well with the objective trajectory ball for each position of the ball. However, for subjects with SZ, the gaze trajectory ball does not overlap as well with the objective trajectory ball.

Different gaze and target ball sizes were tested to determine the most effective combination for clinical evaluation. Table 1 shows that smaller ball sizes for both gaze and objective measures are more effective in distinguishing between groups (*p*-value < 0.05). This suggests that smaller balls provide better characterization and separability of the groups. As the size of both balls increases, there tends to be complete overlap between the gaze and objective balls, reducing their effectiveness in distinguishing between groups. An interesting observation is that when the gaze ball is larger than the target ball, the index tends to decrease. This decrease occurs because the overlap is normalized with respect to the size of the gaze ball, which affects the index value.

### 4.2. Direction

Figure 4 shows the probability density functions in the polar plane that illustrate the distribution of $\left|\frac{1}{N}\sum_{t=1}^{N}\exp^{i(\theta_{o}\left(t\right)-\theta_{g}\left(t\right))}\right|$ for the target trajectory directions (empty bars) and the gaze trajectory directions (filled bars). The top panel displays the probability density functions for the subjects with SZ, while lower panel displays the probability density functions for the control subjects; the angles were computed between consecutive samples. The probability density functions for the subjects with SZ show a tendency towards uniformity, indicating a deviation from the target direction. On the other hand, the control groups present the probability density functions similar to the target ones. Despite the tracking not being perfect, there is a significant similarity between the target and gaze probability density functions. Note that target the probability density functions may show bars in different directions due to the target changing direction each time it reaches a boundary.

Table 1 shows the mean differences in direction between the rectilinear trajectories and the calculated direction index. Statistically significant greater deviation was found between the SZ and the CNT group for both $\Delta \theta_{sample}$ and $\Delta \theta_{regr}$ (0.66 vs. 0.56 for the $\Delta \theta_{sample}$ and 0.31 vs. 0.09 for the $\Delta \theta_{regr}$, respectively; *p* < 0.05).

### 4.3. Area of Polygons

Figure 5 shows the polygons formed for the subjects with SZ (top panel) and for the control subjects (bottom panel). For illustration, three different polygons of the trajectory are presented. In general, it can be observed that most of the subjects with SZ generate polygons with a larger area. This suggests that consecutive points do not form a straight line. The values of the average areas of the polygons calculated with different numbers of points are shown in Table 1. Here, 250 ms corresponds to *n* = 15 points, since we have a sampling frequency of 60 Hz. The value is presented in normalized units (NUs), that is, the area of each segment (for a given *n*) was divided by the maximum area (from SZ and control) and then the average area of the normalized segments was calculated. As can be seen from Table 1, the mean area of the polygons is lower in the control group than in the SZ group, but no statistical differences were found.

### 4.4. Effect Size Analysis

Finally, the observed Hedges’ g values ranged between 0.4 and 0.8, indicating moderate to large effect sizes across the comparisons. These values suggest that the differences between the groups were not only statistically significant but also practically meaningful, reflecting a noticeable shift in the outcome variable. Specifically, a Hedges’ g around 0.4 represents a moderate effect, while values approaching 0.8 indicate a strong effect, implying a substantial separation between group means relative to the pooled variability. This reinforces the relevance of the findings beyond mere statistical significance.

#### Classification Performance

Nowadays, it is of fundamental importance to generate methods to classify a health condition based on bio-physical signals. To cope with this task, assume that we have *m* bio-physical measures $x_{i}$ with *i* = 1, 2, …, *m* related to a single subject, and a feature vector $x={\left[x_{1},x_{2},\ldots ,x_{m}\right]}^{T}$ is constructed, where *T* is the transposition operator. To classify between control and SZ groups, a model must be defined and it should be able to learn the variations found in the feature vectors between samples of each group and label a new vector according to the group. In this work, the feature vectors are composed of the colocalities, area of polygons, and direction indexes. The selected model was the support vector machine; it is often used in biomedical research to classify healthy against pathologic conditions. Since the number of samples in each group is reduced, Leave-One-Out Cross Validation was used (LOOCV) to train and test the SVM classifier. Here, the SVM with a Gaussian kernel was selected. No normalization was applied to the data since the indexes range between zero and one. The SVM model depends on two hyperparameters, sigma and softmargin, which were determined using grid searching. Performance measures of accuracy, sensitivity, and specificity were calculated and achieved values around 90%. The SVM model depends on two hyperparameters, sigma and softmargin, these were optimized using a grid search approach. The search explored a 50 × 50 grid of points, with both hyperparameters ranging from 0.1 to 5, and a step size of 0.1 between consecutive points. The optimal combination found was a softmargin of 0.30 and a sigma value of 1.1.

Permutation Testing was used to assess whether the observed classifier performance is significantly better than chance by comparing it against a null distribution obtained through label permutations. Let $D={\left\{\left(x_{i},y_{i}\right)\right\}}_{i=1}^{N}$ be the dataset of *N* labeled samples, with features $x_{i}\in R^{m}$ and labels $y_{i}\in Y$; A be the classification algorithm SVM yielding a classifier f:R^m^→Y; and ${\hat{\Theta }}_{true}$ be the classification accuracy performance metric estimated via cross validation.

The null hypothesis H_0_ states that there is no relationship between $x_{i}$ and $y_{i}$; thus, any classification performance arises by chance. To evaluate this, we computed the observed statistic: $\hat{\Theta }true$ = Accuracy(A,D), and for b = 1, …, B (B = 1000 permutations):A random permutation was generated $\pi_{b}$ of the label indices {1,…,N}.The permuted dataset was constructed: $D_{b}=\left\{\left(x_{i},y_{\pi_{b(i)}}\right)\right\}$.The classifier was trained and evaluated on $D_{b}$ to obtain: ${\hat{\Theta }}_{b}$ = Accuracy(A,$D_{b}$).

Finally, the empirical estimation of the *p*-value was performed as $p=\frac{1+\sum_{b=1}^{B}I\left(\hat{\Theta }b\geq \hat{\Theta }true\right)}{1+B}$, where I(·) is the indicator function with a value of 1 if the condition A is true and 0 if the condition A is false. Thus, a small *p*-value (*p* < 0.05) indicates that the observed classifier performance is unlikely under the null hypothesis and therefore statistically significant. The test gave a result of *p* < 0.05, indicating the rejection of H_0_.

Bootstrap resampling was used to assess the variability and confidence intervals of classifier performance by resampling from the empirical distribution. It is assumed that the empirical distribution approximates the true population distribution. Thus, for b = 1, …, B (B = 1000 bootstrap resamples), there were sampled *N* instances with a replacement from D to form the bootstrap dataset $D_{b}^{*}={\left\{\left(x_{i}^{*},y_{i}^{*}\right)\right\}}_{i=1}^{N}$ and the accuracy was computed as a performance metric on $D_{b}^{*}$: ${\hat{\Theta }}_{b}^{*}$ = Accuracy(A,$D_{b}^{*}$). After that, the bootstrap mean was $\bar{\Theta }=\frac{1}{B}\sum_{b=1}^{B}{\hat{\Theta }}_{b}^{*}$ and the CI was 95% = [${\hat{\Theta }}_{0.025}$,${\hat{\Theta }}_{0.975}$], where ${\hat{\Theta }}_{p}$ denotes the *p*-th empirical quantile of the set $\{{\hat{\Theta }}_{b}^{*}$}. The performance metrics presented mean values of 90% accuracy, 97% sensitivity, and 93% specificity, with corresponding confidence intervals (CIs) of [0.78, 1.00], [0.78, 1.00], and [0.67, 1.00], respectively.

Finally, Figure 6 presents the feature space where the red balls are the control participants. The gray manifold shows the decision boundary based on the best hyperparameter for an SVM model.

## 5. Discussion

The current pilot study shows statistically significant differences in SPEM metrics between patients with schizophrenia and healthy controls, introducing a quantitative, low-cost method for evaluating SPEM in schizophrenia. The innovative approach of this study lies in the proposal to quantify smooth pursuit through trajectory analysis, focusing on the area of polygons, colocalities, and direction. The main contributions of our study include (a) the evaluation of the fast test based on eye-tracking technology to quantitatively assess clinical features such as processing speed, attention, and visual memory, (b) the introduction of a Data Acquisition System (DAS) designed to record gaze trajectories that are subsequently used to compute spatiotemporal characteristics, and (c) the computed indexes that showed the potential for clinical application to follow-up with subjects with SZ.

Multiple studies including meta-analytic data are consistent, showing that global measures of smooth pursuit eye movement, particularly maintenance gain and leading saccades, are highly discriminative and reliable neurocognitive markers in SZ. Particularly, O’Driscoll and Callahan reported in their meta-analytic review Cohen’s d values of 1.06 (95% CI: 0.92–1.20) and 0.96 (95% CI: 0.81–1.11) for maintenance gain and leading saccades, respectively, of which both effect sizes are considered large by conventional standards [15,16,17,18,29]. Dang et al. [30] employed a ResNet-based attention network (RAnet-ET) to classify schizophrenia using multimodal eye-tracking data, achieving over 96% diagnostic accuracy when combining multiple complex experiments. This is in line with the present data where 94.4% accuracy was observed. Moreover, the simplicity and portability of the presented eye-tracking devices in this study open the door for broader clinical use, including outpatient settings and Telepsychiatry. Incorporating smooth pursuit analysis as part of routine cognitive assessment could assist clinicians in evaluating the efficacy of therapeutic interventions, particularly antipsychotic medications, which are known to influence oculomotor performance. Traditional clinical evaluations rely heavily on subjective symptom reporting and clinician interpretation, introducing variability that can impact diagnostic accuracy and treatment monitoring. The objective nature of eye movement measurements provides a standardized approach that could enhance diagnostic precision and reduce assessment time.

Table 2 summarizes the main characteristics of the studies related to SZ analysis with eye tracking. Most of the studies focus on assessing gaze behavior during static stimuli with high-resolution systems. Consequently, there is a noticeable gap in exploring smooth pursuit in patients with SZ. Smooth pursuit movement, despite its significance in clinical assessments, has received little attention in the literature. This is due to the challenges of synchronizing gaze pattern recordings with smooth pursuit stimuli, as well as the scarcity of protocols and software dedicated to quantifying such movements. Consequently, most of the existing literature has focused on fixations and saccadic movements (as shown in Table 2), neglecting the valuable insights provided by smooth pursuit. Recent studies have renewed interest in SPEM as a marker for psychotic disorders. Lyu et al. [31] evaluated eye movement tasks, including smooth pursuit, and abnormalities have been observed in patients with schizophrenia; their classification model effectively discriminated patients from controls. These abnormalities appear early in the disorder’s progression and may serve as potential markers for schizophrenia. Additionally, Bolazani et.al [32] used eye tracker technology in patients with schizophrenia and bipolar disorder, and their results showed significantly impaired smooth eye movements in both groups compared to controls. Additionally, Meyhoefer et.al [16] analyzed smooth pursuit eye movement data from a large sample of patients with psychosis and healthy controls using multivariate pattern analysis. They obtained balanced accuracies around 60-70% for the prediction of psychosis, psychotic and non-psychotic bipolar disorder, and non-psychotic affective disorder.

This study reinforces the importance of smooth pursuit as a marker for gaze stability, building on prior work that links these oculomotor behaviors to cognitive function. In schizophrenia, previous research has consistently shown that patients struggle to maintain a focused gaze, leading to increased saccadic activity and fragmented gaze trajectories. These disruptions, often visualized as polygonal patterns rather than smooth lines, highlight the complexity of gaze control in schizophrenia and underscore the need for further research into the underlying cognitive and neural mechanisms.

Our automated assessment approach offers a standardized and objective methodology for evaluating gaze abnormalities in schizophrenia, with the potential to reduce clinician workload and enhance diagnostic reliability. By leveraging robust feature engineering, our model achieved high classification accuracy, confirming that the selected features—each capturing unique aspects of gaze dynamics—are both complementary and essential for precise assessment. This multifaceted approach not only improves diagnostic precision but also lays the groundwork for deeper exploration of the cognitive and neurological processes governing gaze behavior.

A key strength of our work is the development of a user-friendly, portable interface tailored for clinical use, with potential applications beyond schizophrenia, including ADHD and Parkinson’s disease. While our findings are promising, the study’s limited sample size highlights the need for larger, more diverse cohorts to validate the generalizability of our approach. Future research should focus on expanding the system’s capabilities to assess a broader range of cognitive traits and clinical conditions, ultimately supporting more personalized and effective interventions in mental health care.

While the EyeTribe Tracker offers accessibility and ease of integration, it presents notable limitations: its low sampling rate (30 Hz) restricts fine-grained temporal analysis of fast eye movements, and it is more sensitive to head movements and signal loss, especially in clinical populations or participants wearing glasses. Despite these constraints, the system proved sufficient for reliably capturing smooth pursuit trajectories in a controlled setting, supporting its potential for use in exploratory clinical eye-tracking studies.

The polygon area did not reveal statistically significant differences between groups. This result suggests that, in the current sample and task configuration, polygon area alone may have limited discriminative value. It is possible that this measure is more sensitive to specific task designs or larger variations in gaze behavior than those captured in our study. Additionally, the relatively small sample size and inter-individual variability may have further reduced the statistical power to detect subtle group-level differences. As such, the limited contribution of the polygon area should be considered a methodological limitation. Nonetheless, we chose to retain this feature in our classification pipeline to explore its potential utility in multivariate contexts, particularly when combined with other trajectory-based features. Future studies with larger samples and refined task paradigms may help clarify the conditions under which the polygon area becomes a more robust marker for distinguishing clinical from control populations.

The small sample size in each group imposes several critical limitations on the reliability and generalizability of the statistical and machine learning analyses: With only nine samples per group, the ability to detect true effects or differences between groups is significantly reduced. This increases the risk of Type II errors, especially for moderate or subtle effects. In addition, machine learning models, including support vector machines, can produce unstable or high-variance estimates when trained on very small datasets. The decision boundary may be overly sensitive to individual samples, which compromises its robustness and reproducibility. Furthermore, small sample sizes increase the risk of overfitting, where the model captures noise of the training set rather than generalizable patterns. In addition, performance metrics like accuracy, sensitivity, specificity, and even confidence intervals derived from bootstrapping may exhibit high variability. Thus, these limitations require cautious interpretation, and the results should be viewed as exploratory or preliminary, and further validation on larger, independent datasets is needed.

The proposed methodology was applied to subjects who were unable to calibrate or perform the activity. This difficulty or inability among some patients with schizophrenia to complete the test suggests that, in certain cases, therapy may not effectively improve cognitive and visual performance; however, the eye-tracking system helps to identify such cases. Finally, antipsychotic medications can significantly affect smooth pursuit eye movements (SPEMs) [4,9,11]. However, the sedative effects of antipsychotics may also impair SPEM by slowing reaction times and reducing overall motor coordination [11,12]. Additionally, individual variability in drug metabolism and dosage can lead to differing effects on eye movement performance. Further research is needed to disentangle the specific contributions of medication effects from the underlying neurobiological deficits in schizophrenia. However, for this study, an assessment of motor abnormalities potentially related to medication side effects was conducted, and none of the patients exhibited clinically evident symptoms. As all the participants were receiving atypical antipsychotics, which are associated with a lower risk of motor side effects compared to typical antipsychotics, the likelihood of such effects influencing the results is considered minimal.

## 6. Conclusions

In conclusion, our study analyzed gaze behavior among individuals with schizophrenia and controls, with a particular focus on their responses during smooth pursuit stimuli. By using spatiotemporal indices of the gaze trajectory, we were able to quantify discernible differences between these groups. Our methodology is practical and accessible, offering a user-friendly approach tailored for clinicians without the need for complex sensor connections. Furthermore, the efficiency of the short task makes it a valuable tool for rapid clinical assessment, enabling timely interventions and informed decision-making. Given the versatility of our approach, it will allow for analysis extension to a range of mental illnesses beyond schizophrenia. Our analytic framework establishes a crucial connection between physiological behavior and cognitive processes. This provides a foundation for future studies to investigate and comprehend a wider range of mental health conditions. Through ongoing research and refinement, we expect our methodology to make a significant contribution to the advancement of our understanding of cognitive function and behavior in various clinical populations. This will ultimately promote improved diagnostic and therapeutic strategies in mental healthcare.

## Figures and Tables

**Figure 1 sensors-25-05212-f001:**
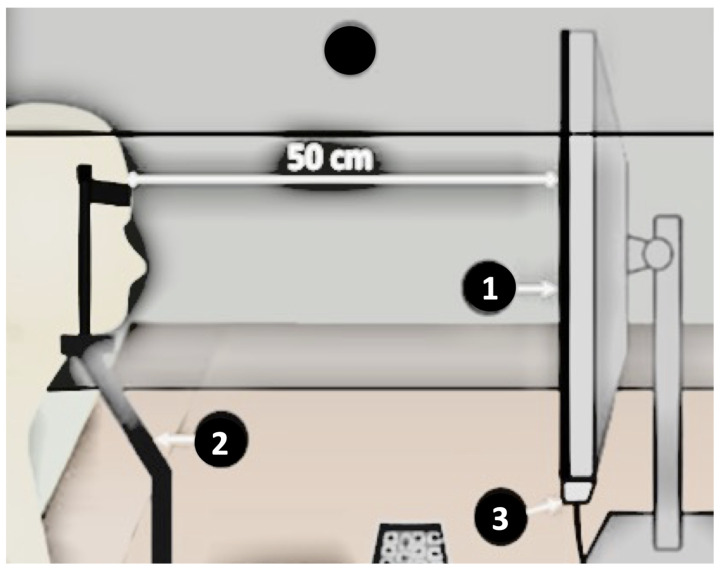
Diagram of the used system for eye-tracking acquisition. (1) Display, (2) chin rest, and (3) eye tracker.

**Figure 2 sensors-25-05212-f002:**
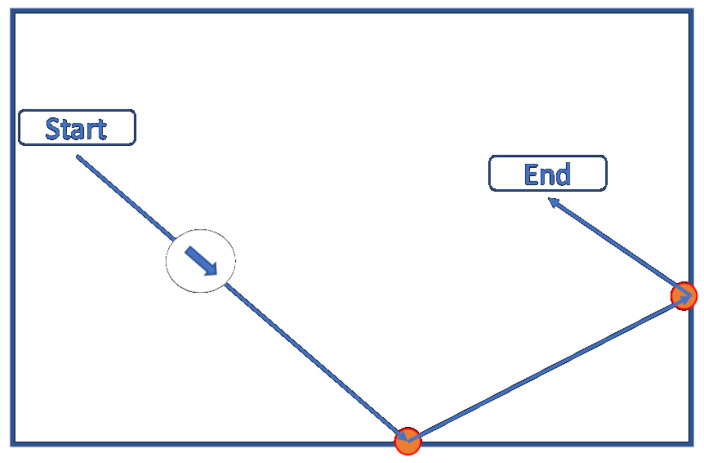
Illustration of a target trajectory. The target direction is indicated by the arrow inside the circle.

**Figure 3 sensors-25-05212-f003:**
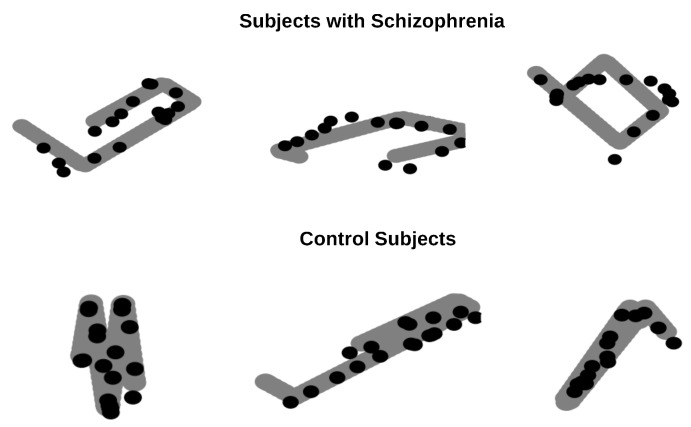
Illustration of the gaze and target trajectories of the spheres used to calculate the colocalization index. The **top** panel shows the trajectories of the subjects with schizophrenia, while the bottom panel shows the trajectories of the control subjects. The filled gray sphere represents the target, and the unfilled black sphere represents the gaze. Please note that the number of samples was selected to facilitate clearer visualization of the trajectories.

**Figure 4 sensors-25-05212-f004:**
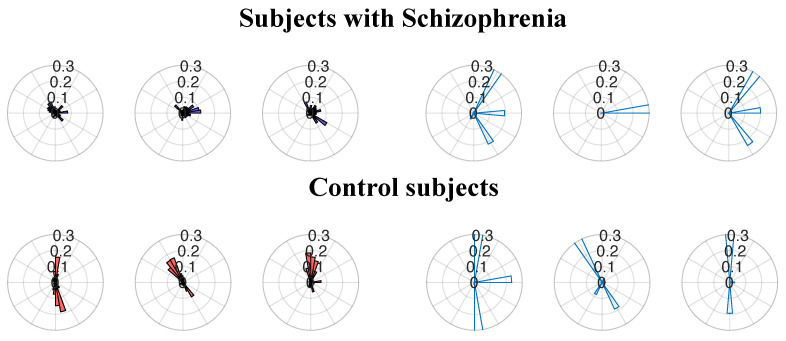
Example of the probability density functions for the gaze and target direction index. The top panel displays the probability density functions of the subjects with SZ, while the **bottom** panel displays the probability density functions of the control subjects. The filled red bars represent the gaze, and the empty blue bars represent the target.

**Figure 5 sensors-25-05212-f005:**
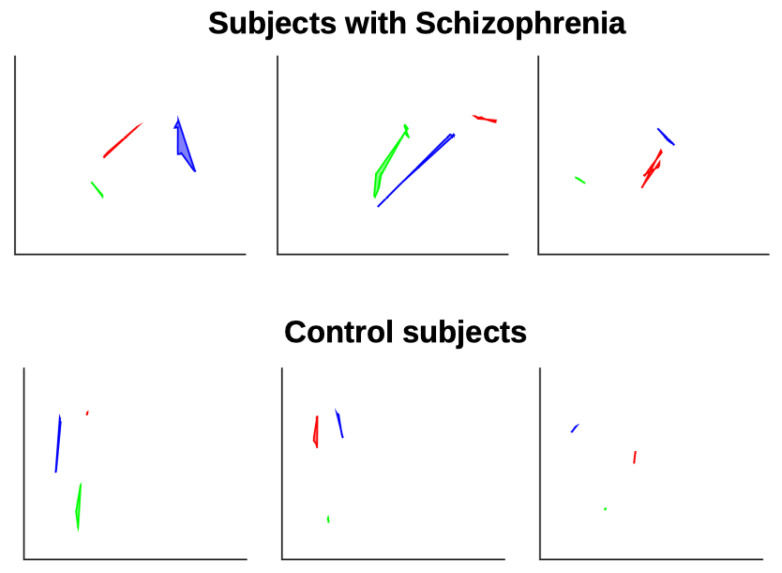
Illustration of the gaze and target polygons used to calculate the area of polygon index. The **top** panel shows the polygons of the subjects with SZ, while the **bottom** panel displays the polygons of the control subjects. Each colored polygon represents a different time in the trajectory.

**Figure 6 sensors-25-05212-f006:**
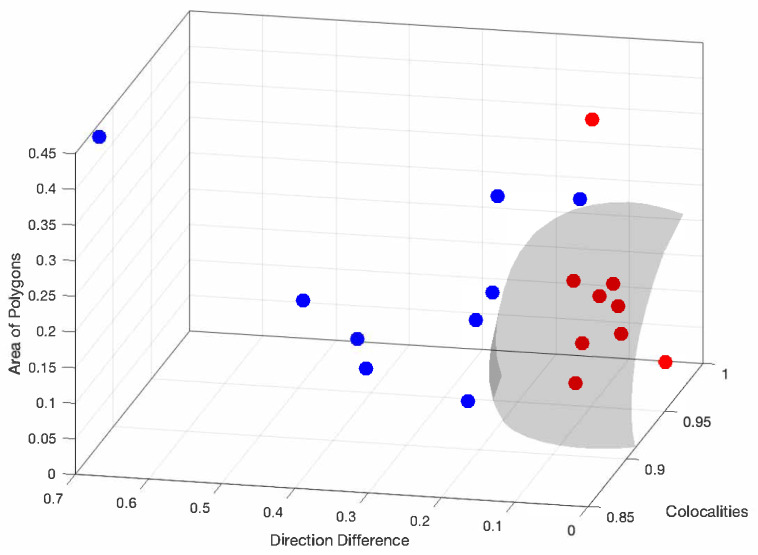
The feature space for evolving regions, angle difference, and co-occurrence, where the blue balls are the participants with SZ and the red balls are the control participants. The gray surface is the discriminant function.

**Table 1 sensors-25-05212-t001:** Mean values of the colocalities, direction, and area of polygon indices between the control (CNT) and subjects with schizophrenia (SZ) during smooth pursuit tracking.

Colocalities Index (Co)																				
	r1=5mm							r1=10mm							r1=15mm					
	r2=2mm		r2=4mm		r2=8mm			r2=2mm		r2=4mm		r2=8mm			r2=2mm		r2=4mm		r2=8mm	
**Subj**	**SZ**	**CNT**	**SZ**	**CNT**	**SZ**	**CNT**		**SZ**	**CNT**	**SZ**	**CNT**	**SZ**	**CNT**		**SZ**	**CNT**	**SZ**	**CNT**	**SZ**	**CNT**
1	0.98	0.99	0.95	0.97	0.39	0.39		1.00	1.00	1.00	1.00	0.99	0.99		1.00	1.00	1.00	1.00	1.00	1.00
2	0.92	0.98	0.88	0.96	0.38	0.39		0.99	1.00	0.99	1.00	0.97	0.99		1.00	1.00	1.00	1.00	1.00	1.00
3	0.96	1.00	0.94	0.97	0.39	0.39		0.99	1.00	0.99	1.00	0.98	1.00		0.99	1.00	1.00	1.00	1.00	1.00
4	0.89	0.98	0.87	0.97	0.39	0.39		1.00	1.00	1.00	1.00	0.96	0.99		1.00	1.00	1.00	1.00	1.00	1.00
5	0.89	0.96	0.88	0.93	0.39	0.39		0.95	1.00	0.96	1.00	0.96	0.98		0.98	1.00	0.98	1.00	0.99	1.00
6	0.35	0.98	0.37	0.95	0.23	0.39		0.61	1.00	0.66	1.00	0.64	0.99		0.73	1.00	0.78	1.00	0.79	1.00
7	0.96	0.99	0.91	0.96	0.38	0.39		1.00	1.00	1.00	1.00	0.98	0.99		1.00	1.00	1.00	1.00	1.00	1.00
8	0.98	0.98	0.97	0.96	0.39	0.39		0.99	1.00	1.00	1.00	0.99	0.99		1.00	1.00	1.00	1.00	1.00	1.00
9	0.93	0.99	0.90	0.98	0.38	0.39		1.00	1.00	1.00	1.00	0.98	1.00		1.00	1.00	1.00	1.00	1.00	1.00
mean	0.87 *	0.98	0.85 *	0.96	0.37	0.39		0.95	1.00	0.95	1.00	0.94 *	0.99		0.97	1.00	0.97	1.00	0.97	1.00
	**Directions (rad)**										**Area of Polygons (NU) (A)**									
	$\Delta \theta_{\mathrm{sample}}$		$\Delta \theta_{\mathrm{regr}}$								**250 ms**		**500 ms**		**750 ms**		**1000 ms**		**1500 ms**	
	**SZ**	**CNT**	**SZ**	**CNT**							**SZ**	**CNT**	**SZ**	**CNT**	**SZ**	**CNT**	**SZ**	**CNT**	**SZ**	**CNT**
1	0.61	0.43	0.49	0.02							0.00	0.03	0.00	0.09	0.01	0.10	0.00	0.04	0.13	0.04
2	0.57	0.49	0.70	0.07							0.01	0.00	0.08	0.02	0.17	0.06	0.13	0.00	0.43	0.09
3	0.68	0.51	0.11	0.15							0.20	0.01	0.20	0.01	0.16	0.02	0.03	0.02	0.30	0.14
4	0.60	0.55	0.19	0.08							0.01	0.00	0.11	0.01	0.04	0.06	0.02	0.07	0.10	0.12
5	0.62	0.55	0.16	0.10							0.06	0.04	0.09	0.06	0.22	0.08	0.35	0.06	0.38	0.06
6	0.89	0.64	0.33	0.10							0.20	0.27	0.49	0.33	0.49	0.31	0.19	0.24	0.19	0.40
7	0.61	0.68	0.37	0.10							0.01	0.00	0.03	0.00	0.06	0.00	0.04	0.01	0.08	0.14
8	0.58	0.49	0.25	0.08							0.00	0.01	0.03	0.02	0.07	0.02	0.02	0.02	0.13	0.16
9	0.62	0.71	0.20	0.14							0.33	0.01	0.34	0.04	0.34	0.06	0.13	0.02	0.19	0.05
mean	0.66 *	0.56	0.31 *	0.09							0.09	0.04	0.15	0.06	0.17	0.08	**0.10**	0.06	0.21	0.13

* Statistical differences between CNT and SZ at specific index parameter. *Sample* is the mean difference direction computed at each sample. *Regr* is the direction of the mean difference calculated with the slopes obtained with the points between the bounces of the balls. NU stands for normalize unit. r1 stands for ball radius of the object and r2 is for ball radius of the gaze. All the indexes show a Hedges’ g ≥ 0.4.

**Table 2 sensors-25-05212-t002:** Summary of the main characteristic of the studies related to SZ analysis using eye trackers. DSS: dynamic or static stimuli, Sac: saccadic, SP: smooth pursuit, AOI: Area of Interest, SZ: number of subjects with schizophrenia, and CNT: number of control subjects.

Reference	SZ/CNT	Eye Tracker Resolution	DSS	Fix	Sac	SP	AOI
[6]	85/252	High	E, D	∘		∘	
[15]	21/38	Low	E	∘			∘
[5]	23/23	High	E	∘			∘
[7]	32/33	High	E	∘	∘		
[9]	32/37	High	E	∘			
[33]	20/20	High	E	∘	∘		

## Data Availability

The data are not publicly available due to legal and ethical restrictions. Any requests for access to the data must be submitted to the corresponding author and subsequently to the ethics committee for review and approval before distribution.

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
