# Peer review of "Assessing Smooth Pursuit Eye Movements Using Eye-Tracking Technology in Patients with Schizophrenia Under Treatment: A Pilot Study"

_sensors, 2025, doi:10.3390/s25165212_

Round 1

Reviewer 1 Report

Comments and Suggestions for Authors

The article requires revision. The results of the study are questionable for several reasons.
This is too small a number of participants. Schizophrenia is a very polysymptomatic disease. It is not specified which patients participated in the study, with what form of schizophrenia. The average age of patients and the average age of the control group are not specified. It is not clear whether they differed in age or not. It is not specified what therapy the patients received and the dosage.
The authors make mistakes in presenting ideas about eye movements. The introduction requires revision, since it does not provide an idea of the study of eye movements in schizophrenia. Information about this appears in the discussion. Data from eye movement studies, including tracking eye movements, is quite sufficient. The goal statement should reflect the state of the problem and what the authors want to add to its solution.

Author Response

Response Letter.

To Reviewer # 1. We would like to express our gratitude for your valuable feedback and suggestions for improving our manuscript. We have carefully reviewed all your comments and have made the necessary revisions accordingly.

Comments 1: This is too small a number of participants. Schizophrenia is a very polysymptomatic disease. It is not specified which patients participated in the study, with what form of schizophrenia.

Response 1: We thank the reviewer for this observation. To address the comment, we have included the following lines in the manuscript to provide the requested information:

... nine individuals diagnosed with paranoid schizophrenia (SZ group) undergoing treatment with atypical antipsychotics (risperidone (2 mg), olanzapine (10–15 mg), quetiapine (350–650 mg) for at least 2 weeks

Regarding the small number of participants, we appreciate the reviewer's important question. The primary objective was exploratory and focused on the evaluation and validation of a novel eye-tracking methodology and system developed specifically for people diagnosed with schizophrenia.

This initial phase of the research was designed to:

  1. Assess the technical performance and robustness of the eye-tracking setup and software in a clinical context;
  2. Validate the experimental paradigm (such as the target trajectory and task design) for its effectiveness in eliciting and capturing reliable gaze data;

 Comments 2:  The average age of patients and the average age of the control group are not specified. It is not clear whether they differed in age or not.

Response 2: We thank to the reviewer for this observation.  To add the missing information, the following text was added to the article.

The age of the SZ group is 24.7 7.5 years. The age of the CNT group is 22 3.2 years.

Comments 3: It is not specified what therapy the patients received and the dosage.  

Response 3: We thank to the reviewer for this comment. We have included the following lines in the manuscript. They provide the requested information.

... nine individuals diagnosed with paranoid schizophrenia (SZ group) undergoing treatment with atypical antipsychotics (risperidone (2 mg), olanzapine (10–15 mg), quetiapine (350–650 mg) for at least 2 weeks

Coments 4: The authors make mistakes in presenting ideas about eye movements. The introduction requires revision, since it does not provide an idea of the study of eye movements in schizophrenia. Information about this appears in the discussion.  

Response 4: Many thanks for this comment. After reading the introduction with caution we recognize some inaccurate sentences which could be misunderstood by the reader. For this reason we decided to rewrite some parts of the introduction eliminating inaccuracies. The Introduction should be changed as follows:

The diagnosis of SZ is based on the detection of a minimum number of key alterations in mental function [4] divided into cognitive domains: processing speed, attention/vigilance, working memory, social cognition, reasoning and problem solving, learning and visual memory. In severe forms, the evident alterations allow a faster diagnosis. However, detecting SZ in its early stages is challenging as symptoms may not be clear, and an objective, quantitative evaluation strategy is lacking. Additionally regular assessments after diagnosis are crucial to evaluate the patient’s evolution under medication. This evaluation demands expert personnel employing quantitative measures to determine therapy success. Therefore, finding innovative strategies to aid physicians in diagnosing, following-up and detecting early stages of SZ is imperative.

Previously published works have shown that individuals with SZ exhibit significant alterations in various domains of visual processing, particularly in perceptual organization and motion detection, which are consistently impaired in this population [5–7]. These visual processing deficits are not isolated phenomena but are closely intertwined with broader cognitive dysfunctions. For instance, impairments in perceptual organization have been linked to deficits in attention and working memory [8,9]. At the same time, abnormalities in eye movements, particularly in smooth pursuit, reflect underlying disruptions in processing speed and visual memory [10,11]. Since gaze control is a cognitive-motor function, disruptions in visual tracking tasks can serve as indirect markers of attentional lapses and slowed cognitive processing [2,12]. This underscores the importance of studying smooth pursuit eye movements as to assess cognitive impairments characteristics of SZ, specifically those related to attention, processing speed, and visual memory. In addition, SZ patients consistently show increased saccade frequency, prolonged fixation durations, and reduced reading fluency, reflecting both oculomotor and higher-level semantic processing deficits. Furthermore, individuals with SZ show impaired top-down attentional control, leading to increased fixations and revisits to salient but irrelevant stimuli, and difficulty suppressing distractors [17 ,33].  In particular, abnormalities in smooth pursuit have been shown to serve as indicators of psychosis in disorders such as schizophrenia, schizoaffective disorder, and psychotic bipolar disorder [14,19–21].  While previous studies have examined eye movement abnormalities in schizophrenia, such research have largely been restricted to controlled experimental settings due to the challenge of obtaining reliable gaze data. Smooth pursuit eye movements, in particular, are recognized as potential biomarkers of schizophrenia, yet their clinical use has been limited by the technical difficulties in accurately recording and analyzing these motions.

Recent studies have increasingly employed eye-tracking technology to investigate smooth pursuit eye movement (SPEM) abnormalities in schizophrenia, highlighting its potential as a biomarker for the disorder. For instance, Ales et al. demonstrated that individuals instructed to feign schizophrenia could not replicate the SPEM deficits observed in actual patients, underscoring the specificity of these oculomotor anomalies to the disorder [18].  Similarly, Komogortsev and Karpov developed a ternary classification system to distinguish between fixations, saccades, and smooth pursuits, demonstrating the feasibility of automated SPEM analysis in clinical populations [34]. These findings align with our study’s approach, which leverages quantitative gaze trajectory features to distinguish between medicated schizophrenia patients and healthy controls. Our methodology builds upon this foundation by integrating machine learning classification, offering a novel contribution to the growing body of literature on SPEM-based diagnostics.  Furthermore, recently published reviews have highlighted the diagnostic potential of Smooth Pursuit Eye Movements (SPEM). Lima and Ventura reviewed psychophysical eye-tracking designs and emphasized the utility of smooth pursuit metrics in assessing perceptual and cognitive déficits [35]. Startsev et al. review eye movement detection research, suggesting that SPEM abnormalities may serve as reliable indicators of cognitive dysfunction and disease progression [36].

 The goal of our study is to bridge the gap of eye movements tracking in clinical settings  by conducting a comparative analysis of smooth pursuit eye movements in medicated individuals with schizophrenia versus healthy controls, using a visual tracking task. We aim to demonstrate that quantitative metrics (specifically, trajectory-based measures such as co-localities, directionality, and polygonal area) can be used effectively with modern eye-tracking technology. Furthermore, by developing a Support Vector Machine (SVM) classifier, we seek to establish a framework for an objective, clinically valuable tool to assess gaze behavior and aid in the diagnosis and monitoring of schizophrenia.

.

Comments 5: Data from eye movement studies, including tracking eye movements, is quite sufficient. The goal statement should reflect the state of the problem and what the authors want to add to its solution.

Response 5:  We thank the reviewer for this observation. We agree that there are numerous contributions in the field of eye-tracking; however, only a few specifically address our area of focus, as cited in the Introduction and Discussion. We have revised the goal statement of our work to better reflect both the underlying problem and our intended contribution:

The goal of our study is to bridge this gap by conducting a comparative analysis of smooth pursuit eye movements in medicated individuals with schizophrenia versus healthy controls, using a visual tracking task. We aim to demonstrate that quantitative metrics (specifically, trajectory-based measures such as co-localities, directionality, and polygonal area) can be used effectively with modern eye-tracking technology. Furthermore, by developing a Support Vector Machine (SVM) classifier, we seek to establish a framework for an objective, clinically valuable tool to assess gaze behavior and aid in the diagnosis and monitoring of schizophrenia.

Reviewer 2 Report

Comments and Suggestions for Authors

The following improvements could be made to the article:

  1. The connection between visual alterations and attention, processing speed and visual memory could be made clearer in the introduction with supporting citations.
  2. Provide more details on the olanzapine equivalents that the patients with schizophrenia were treated with. Justify the use of linear interpolation for resampling the eye-tracking data to 60 Hz.
  3. Comparing existing literature with similar eye-tracking methodologies or looking at studies that have investigated smooth pursuit eye movements in schizophrenia in more detail.
  4. Explain the choice of the Wilcoxon–Mann–Whitney test in more detail. Was a non-parametric test chosen because the data violated the normality assumption?
  5. Provide a better understanding of the magnitude of the observed differences by reporting effect sizes alongside p-values.
  6. You should provide specific information about the eye tracker, such as its model number, manufacturer, and limitations
  7. Explain the specific libraries or toolboxes used in Processing 3 for data analysis and visualisation.
  8. Provide a range of tested sigma and soft margin values obtained through grid searching.
  9. Consider and discuss how antipsychotic medication (such as olanzapine) might affect smooth pursuit eye movements.
  10. Clearly state the limitations imposed by the small sample size (n = 9 per group).
  • Figure 2: Add information about the speed of the white ball.
  • Figure 3: It's hard to visually discern the difference in overlap between the SZ and control groups from the figure alone.
  • Figure 4: Clarify in the caption that these are normalised
  • Figure 5: Similar to Figure 3, it isn't easy to assess a consistent difference in polygon area visually

Reviewer 3 Report

Comments and Suggestions for Authors

Major

Line 11–12: Sample size

Issue: Only 9 participants per group (n=18) significantly limits statistical power and generalizability.

Question: Was a power analysis conducted to justify this sample size?

Line 74–77: Control group selection

Issue: Healthy participants are “previously classified as healthy by a psychiatrist” but no further screening (e.g., SCID, MMSE) is described.

Suggestion: Include formal psychiatric screening criteria to ensure valid group classification.

Line 88–90: Eye tracker setup

Concern: EyeTribe’s low sampling rate (20 Hz, interpolated to 60 Hz) is a limitation for studying SPEM.

Question: How do authors justify resampling without introducing artificial smoothness or bias?

Line 157–158: Attention claim

Issue: The statement "failure to follow the moving target accurately could suggest difficulties in directing and maintaining attention" lacks citation and assumes causality.

Question: Could motor or oculomotor abnormalities unrelated to attention explain this deviation?

Line 208: Statistical analysis

Concern: Non-normality was assessed with Kolmogorov-Smirnov, but this test is known to have low sensitivity at small sample sizes.

Suggestion: Consider using Shapiro-Wilk instead and include effect sizes (e.g., Cliff’s delta) alongside p-values.

Line 273–275: Classifier evaluation

Issue: LOOCV with such a small dataset is prone to overfitting.

Question: Were permutation tests or bootstrapping used to assess classifier robustness?

Line 284–285: Accuracy claims

Issue: “Around 90%” classification accuracy is mentioned multiple times.

Clarification needed: What is the exact accuracy, sensitivity, specificity? Are confidence intervals provided?

Minor

Line 30: “Affects approximately 1 in 222 people (0.45%)”

Correction: WHO reports schizophrenia prevalence at ~0.3–0.7%. The number “1 in 222” should be clearly sourced.

Line 97: Calibration task

Suggestion: Please clarify how calibration quality was assessed or rejected for each participant.

Line 214–217: Co-localities results

Question: Was there test–retest reliability or intra-subject variability assessed for this metric?

Line 249–257: Polygon area analysis

Observation: No statistical significance was found; this should be discussed more transparently in the Discussion as a limitation.

Line 289–291: Meta-analysis citation

Suggestion: Include specific effect size values or studies referenced in “Meta-analytic data indicate…” to support claim.

Line 334–339: Gaze fragmentation

Suggestion: Include example figures of raw gaze paths (not just polygon overlays) to visually demonstrate these differences.

Language and Clarity

Title: Capitalize appropriately – “Smooth Pursuit Eye Movements” and “Schizophrenic Patients”.

Abstract Line 5: Rephrase “particularly, smooth pursuit eye movements” → “in particular, smooth pursuit eye movements”.

Throughout: The manuscript would benefit from professional language editing. There are frequent minor grammar issues (e.g., "radio" instead of "radius", "inherently accounts" instead of "inherently account").

Recommendation:

Major Revision

The concept is valuable, and the technical implementation is innovative for low-cost settings. However, methodological clarifications, more rigorous statistics, and clearer writing are required before it can be recommended for publication.

Round 2

Reviewer 2 Report

Comments and Suggestions for Authors

The authors have satisfactorily responded to all my questions and made the necessary changes to the manuscript.

In my opinion, the article could be accepted.

Author Response

Comment 1: The authors have satisfactorily responded to all my questions and made the necessary changes to the manuscript.

Response 1:  We sincerely appreciate the time and attention you took to review our article. Your comments have been invaluable in improving the quality of our work.

Valdemar Emigdio Arce Guevara
On behalf of the authors